# The lncRNA SENCR knockdown alleviates vascular calcification via miR-4731-5p by suppressing endoplasmic reticulum stress

Yongpan Huang[1]◉, Wei Zhan[2]◉, Chong Song[1]◉, Meihua Tan[3], Li Wu[3], Sina Wu[4]*

1 School of Medicine, Changsha Social Work College, Changsha, Hunan, China, 2 Reproductive and Genetic Hospital of CITIC-Xiangya, Changsha, China, 3 Hunan Carnation Endowment Industry Investment Real Estate Co., Ltd, Changsha, China, 4 Department of Respiratory, Affiliated Nanhua Hospital, Hengyang Medical School, University of South China, Hengyang, Hunan, China

◉ These authors contributed equally to this work.
* Wusn_usc@163.com

## Abstract

### Background

Accumulation of calcium phosphate crystals is associated with vascular calcification (VC); however, the mechanism that promotes VC remains unclear. Accumulating evidence indicates that smooth muscle and endothelial cell-enriched migration/differentiation-associated lncRNA (SENCR) exerts a critical role in VC. This work focuses on the molecules involved in β-glycerophosphate-induced osteogenic differentiation of vascular smooth muscle cells (VSMCs) through SENCR epigenetic modification of Runx2 in an endoplasmic reticulum stress (ERS)-dependent manner.

### Methods

We cultured VSMCs to explore the relationship among β-glycerophosphate, SENCR, and VC and also investigate the function of SENCR in β-glycerophosphate-induced osteogenic differentiation and VC in vitro.

### Results

Our findings indicate that β-glycerophosphate enhanced SENCR, MSH homeobox 2, Runx2, ERS-related markers, alkaline phosphatase activity, and cellular calcium deposition and suppressed the expression of α-SMA, SM 22α, and miR-4731-5p. SENCR silencing increased miR-4731-5p expression, which subsequently inhibited β-glycerophosphate-associated endoplasmic reticulum stress at the post-transcriptional level. Critically, the facts that direct interplay between SENCR and miR-4731-5p, and the downregulation of miR-4731-5p efficiently reversed the suppression of ERS-induced by SENCR silencing were observed. Collectively, the present study clarifies a novel mechanism by which downregulation of SRNRC

**Data availability statement:** The relevant data can be found at 10.17605/OSF.IO/4SE8G.

**Funding:** This work was supported by the funding of Professor and PhD at Changsha Social Work College (2023JB24) and the Hunan Provincial Natural Science Foundation (no. 2023JJ60262). The funders supported study design, data collection and analysis, decision to publish, and preparation of the manuscript.

**Competing interests:** All the authors declared no competing interests.

contributes to the ERS-dependent osteogenic differentiation of VSMCs and VC by sponging miR-4731-5p. This study demonstrates that SENCR/miR-4731-5p axis is involved in β-glycerophosphate-mediated VC *in vitro*.

## Introduction

Vascular calcification (VC) refers to the deposition of calcium-phosphate crystals within the intima and media layers of the vasculature and/or heart valves, with high morbidity and mortality in aging individuals and diabetes, cardiovascular diseases, inflammation, and patients with chronic kidney disease (CKD) [1–4]. Emerging evidence suggests that VC is a gene-regulated biological process involving dysregulation of the balance between matrix mineral metabolism and activation of cellular signaling pathways, which are associated with osteogenic differentiation [5,6]. Risk factors that drive the progression of vascular, including inflammation, oxidative stress, and calcium and phosphate imbalances, have been reported to stimulate the development of VC in individuals with CKD and cardiovascular incidents. High blood phosphate levels are a crucial risk factor for cardiovascular morbidity and mortality in CKD patients. In addition, high phosphate levels, which are associated with osteogenic differentiation, are pivotal stimulators of calcification. Suppression of the osteogenic differentiation of vascular smooth muscle cells may be a promising strategy to reverse or retard vascular calcification in patients with CKD. Vascular calcification is characteristic of vascular stiffness in aging individuals. Many studies have indicated that endoplasmic reticulum stress (ERS) is a crucial contributor to the deterioration of vascular calcification [7–9]. The endoplasmic reticulum (ER) is an intracellular membranous organelle that regulates protein synthesis, folding, maturation, and post-translational modifications of secretory and transmembrane proteins. ERS occurs once unfolded/misfolded proteins accumulate through disturbances in Alzheimer disease, diabetes, and CKD, which disturb ER homeostasis in vascular smooth muscle cells (VSMCs) through increasing osteogenic transition.

Long non-coding RNAs (lncRNAs) are known to be vigorous agents, involved in regulating multifaceted aspects of in physiological and pathological functions in body. It has been reported that lncRNAs play multiple roles, including interactions with proteins, interference with microRNAs, modification of the epigenome, regulation of gene expression via recruitment to gene promoters, and participation in RNA shear and modification [10]. A well-known RNA interaction model for post-transcriptional regulation showed that some transcripts, including lncRNAs, may sequester a miRNA and halt its inhibitory function on other mRNA transcripts. Dysregulation of lncRNAs is crucial in vascular calcification. Emerging evidence indicates that smooth muscle and endothelial cell-enriched migration/differentiation-associated lncRNA SENCR is involved in vascular dysfunction and atherosclerosis by modulating vascular smooth muscle cell phenotypes or vascular remodeling [11]. However, whether dysfunction of the SENCR occurs during β-glycerophosphate-induced VC pathogenesis remains largely unknown.

In this study, we predicted potential SENCR/miR-4731-5p pathway that is overwhelmingly likely complicated in VC. We intended to investigate the role of SENCR/miR-4731-5p in VC *in vitro* study using β-glycerophosphate-induced VSMCs model.

## Materials & Methods

### Cell culture

VSMCs were purchased from ScienCell Research Laboratories (Carlsbad, CA). Primary VSMCs between passages three and seven were incubated in a medium containing high-glucose Dulbecco's modified Eagle's medium (DMEM; Gibco, Rockville, MD, USA) supplemented with 10% fetal bovine serum (FBS; Gibco, Rockville, MD, USA), 100 ng/mL fibroblast growth factor cytokine (Peprotech, Rocky Hill, USA), and 1% streptomycin and penicillin (Wisent Biological Products, Canada) at 37 °C with 5% $CO_2$. In view of its ability to enhance VSMCs calcification, β-glycerophosphate at 10 mM (Sigma-Aldrich, St. Louis, MO; CAS:13408-09-8) was used to induce VSMCs calcification. When VSMCs were incubated in DMEM containing 10% FBS and grown to a density of 50%-60%, 2% FBS, 10 mM β-glycerophosphate and 3mM calcium choloride were added to induce VSMCs calcification for 0–10 days. The medium was changed every 2 days. Additionally, the ERS inhibitor 4-phenyl butyric acid (PBA; Sigma-Aldrich, Saint Louis, MO, USA; CAS: 1821-12-1) was added to investigate the role of SENCR in β-glycerophosphate-induced VC. VSMCs were pretreated with 5mM 4-PBA for 2h and then cultured for 7d with or without 10mM β-glycerophosphate.

### Cell migration and wound healing assay

Pretreated cells were seeded in 6-well plates and cultured in DMEM containing 10% FBS until they reached 70% confluence. A pipette tip was used to mark a straight wound in each petri dish. The plate was rinsed with PBS to remove cell debris and then switched to DMEM containing 2% FBS with or without 10 mM β-glycerophosphate to continue the culture. The initial images were photographed and recorded using Image-Pro Plus 4.5 software (Olympus IX71; Tokyo, Japan). The migration distance of cells was evaluated after 24 h.

### Alizarin Red S staining

β-glycerophosphate (10 mM) was used to induce calcification of VSMCs for 0–10 days, and Alizarin Red S staining was performed to verify calcium nodule deposition. VSMCs that reached the observation endpoint were handled with PBS three times and fixed with 4% paraformaldehyde for 30 min at room temperature. After rinsing with PBS, VSMCs were stained with alizarin red S (Sigma-Aldrich, Saint Louis, MO; CAS 130–22.3), and the staining results were collected.

### ALP activity analysis

The ALP activity was detected using a commercial kit (Cat. No. A059-3–1) according to the manufacturer's instructions. The cell layer was cleaved on ice with lysis buffer for 30 min. The lysate was collected and centrifuged at 12,000rpm for 20 minutes at 4 °C. Distilled water was used to zero the spectrophotometer, and the absorbance change was continuously monitored for 1–3 min to compare ALP activity.

### RNA transfection

VSMCs were seeded in a 6-well plate at 37 °C and 5% $CO_2$ overnight. Polybrene was mixed with 2500μL of complete medium to a final concentration of 5 μg/mL. Lentiviral stock solution (30μL) was added to the diluent and gently mixed by pipetting. When siSENCRs were transfected, the medium was discarded, and VSMCs were covered with the virus transfection mixture. A control group (blank or negative control) was established. The cells were incubated overnight at 37 °C with 5% $CO_2$ and removed after 24h. The viral solution was added to a complete culture solution (0.5 mL). The cells were

incubated overnight at 37 °C with 5% $CO_2$. Overexpression lentivirus was transfected as previously described. The miR-4731-5p mimic, mimic control, inhibitor, and inhibitor control were inoculated into 24-well plates at–50–60% density and then transfected with 50nM synthetic oligonucleotides or 2μg vector for 6h at 37°C (Invitrogen; Thermo Fisher Scientific, Inc.) using Lipofectamine® 2000, according to the manufacturer's instructions. After transfection, cells were cultured in fresh DMEM for 24 h or used in subsequent experiments. All RNA sequences were designed and synthesized by Gene-Pharma (Shanghai, China). The target sequence of the SENCR siRNA was as follows: sense 5' GGGCGCAUUGUUAG-GAGAATT 3' antisense 5' UUCUCCUAACAAUGCGCCCTT 3'.

### RNA extraction and quantitative Polymerase Chain Reaction (qRT-PCR)

Total RNA was extracted using an RNA extraction kit (#ER501–01; TransGen). RNA was reverse-transcribed using a ommercialccc kit (Thermo, K1621) in accordance with the manufacturer's instructions to obtain the corresponding cDNA. The mRNA levels were measured using CFX96 real-time PCR detection (Bio-Rad) with SYBR Green PCR core reagents (TaKaRa; #RR820A). Each sample was analyzed in triplicate and the amount of RNA in each sample was normalized using the housekeeping gene GAPDH. Primers used for reverse transcription-quantitative PCR were purchased from Sangon Biotech (Shanghai, China). The primer sequences used were as follows: SENCR F: 5-' GTTACCTTGTCCAC GCTCTCSENCR-3,' R: 5'-GTTTGAAGGTCGGTAGAGCC-3'; GAPDH F: 5'-GGAGTCCACTGGCGTCTTCA-3', R: 5' -GTCATGAGTCCTTCCACGATACC- 3'.

### Western blotting

After the intervention of VSMCs, an appropriate volume of a mixture of RIPA cell lysate (Solarbio, Beijing, China) and PMSF (Solarbio) was added, and the total cell extracts were lysed on ice for 30 min. The supernatant was collected after centrifugation at 12,000rpm at 4 °C for 20 min. The concentrations of the protein samples were determined using a bicinchoninic assay detection kit (Solarbio, Beijing, China) on a microplate reader (Thermo Fisher Scientific, US). Loading buffer was used to unify the total protein amount of each sample (20–40 μg). Total protein was separated using 10% or 12% sodium dodecyl sulfate-polyacrylamide gel electrophoresis and transferred onto a polyvinylidene fluoride membrane (Solarbio, Beijing, China). Membranes were blocked with 5% bovine serum albumin at room temperature for 2 h and then incubated overnight at 4 °C with the following primary antibodies: rabbit anti-Runx2 (1:500; Santa Cruz Biotech-nology, sc-101145), anti-Msx2 (1:2000, Abcam, ab227720), anti-α-SMA (1:1000, Cell Signaling Technology, #19245), anti-SM22α (1:1000, Proteintech, 10493–1-AP), anti-ATF4 (1:500, Proteintech, 10835-I-AP), anti-glucose regulatory protein 78 (GRP78; 1:1000, Proteintech, 11587–1-AP), anti-PERK (1:1000, Cell Signaling Technology, #3192s), and anti-phosphorylated PERK (1:500, Affinity Bioscience, DF75F6). The membrane was washed with TBST and incubated with secondary antibodies conjugated to horseradish peroxidase for 2 h. Western blotting intensity was quantified and scanned using Image Lab software (BioRad, Hercules, CA, USA).

### Statistical analysis

Statistical analyses were performed using GraphPad Prism 7.0 (Inc, La Jolla, CA, USA). Quantitative results are expressed as the mean±standard deviation. The differences between the two groups were analyzed using $t$-test. One-way analysis of variance and Tukey's post hoc test were used to compare multiple groups. Statistical significance was set at $P < 0.05$.

## Results

### Excess β-glycerophosphate promotes VSMCs osteogenic differentiation and VC in VSMCs

To investigate the increase in intracellular calcium nodules in VSMCs treated with β-glycerophosphate, Alizarin Red S staining was utilized on VSMCs. As shown in Fig 1A, VSMCs were treated with β-glycerophosphate or not for 0, 3, 7, or

10 days. The results showed that β-glycerophosphate increased intracellular calcium nodules and ALP activity in VSMCs, particularly on day 7. Furthermore, β-glycerophosphate upregulated SENCR expression in VSMCs in a time-dependent manner (Fig 1C). These findings indicated that SENCR was positively correlated with the progression of β-glycerophosphate-induced VSMCs calcification *in vitro*. VSMCs were treated with β-glycerophosphate for seven days to establish an *in vitro* calcification model.

## β-glycerophosphate-induced phenotypic switching and ERS in VSMCs

Studies have demonstrated ER stress and phenotypic switching of VSMCs in CKD-mediated VC [5], and the levels of the phenotypic and ER stress-related marker GRP78, including the PERK/ATF4 arm of the UPR signal, were assessed. We examined the expression of synthetic and contractile markers, and phenotypic switching to assess whether β-glycerophosphate plays a role in ERS and phenotypic switching. Analysis of the results indicated that the contractile phenotypic marker SM22α was downregulated, whereas the synthetic phenotypic markers Col I, Msx-2, and Runx2 were elevated in calcified VSMCs (Fig 2A, B, C, D). Furthermore, the expression of Msx-2 and Runx2 increased, whereas that of SM22α and α-SMA decreased in calcified VSMCs (Fig 2E). In addition, some hints were exhibited with the facts that ERS-related proteins p-PERK, GRP78, and ATF4 was upregulated (Fig 2F). These findings indicated that β-glycerophosphate accelerates both osteoblast-like differentiation and ERS in VSMCs.

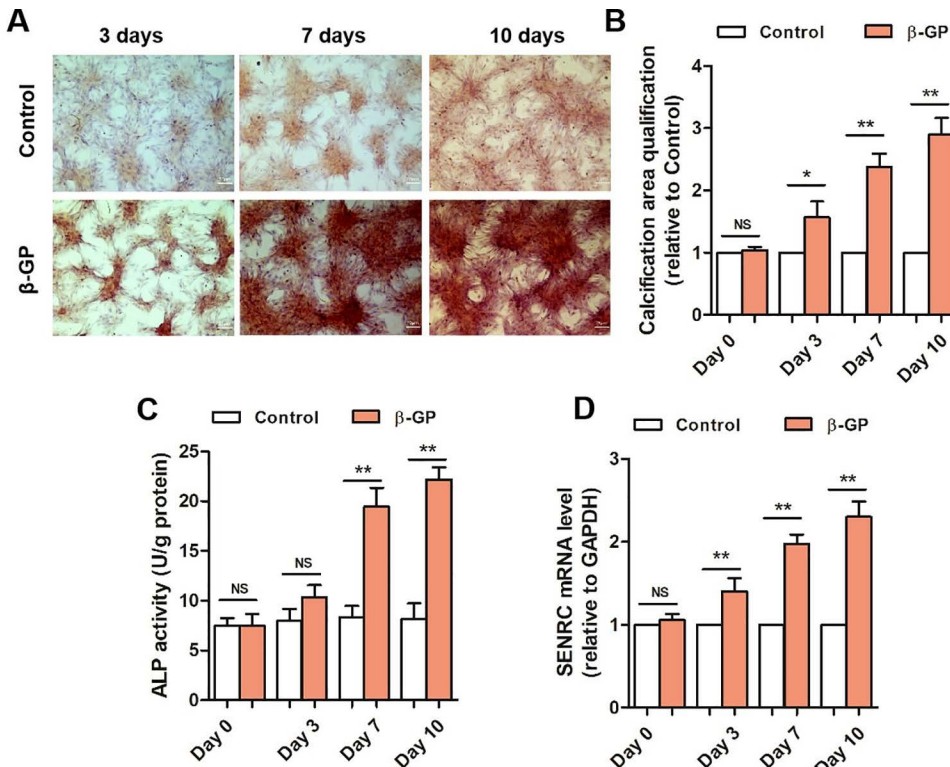

**Fig 1. Excess β-glycerophosphate promotes VSMCs osteogenic differentiation and VC in VSMCs.** A and B. Representative alizarin red S staining images and quantified calcification area demonstrated calcium salt deposition in VSMCs for different duration. C. ALP activity incubated with β-glycerophosphate for 0, 3, 7, 10 days. D. SENRC mRNA expression with β-glycerophosphate-induced calcification was remarkably increased in VSMCs (n=3). One-way analysis of variance and Tukey's post hoc test were used to compare multiple groups. *P<0.05, **P<0.01 vs. Control, NS no significance.

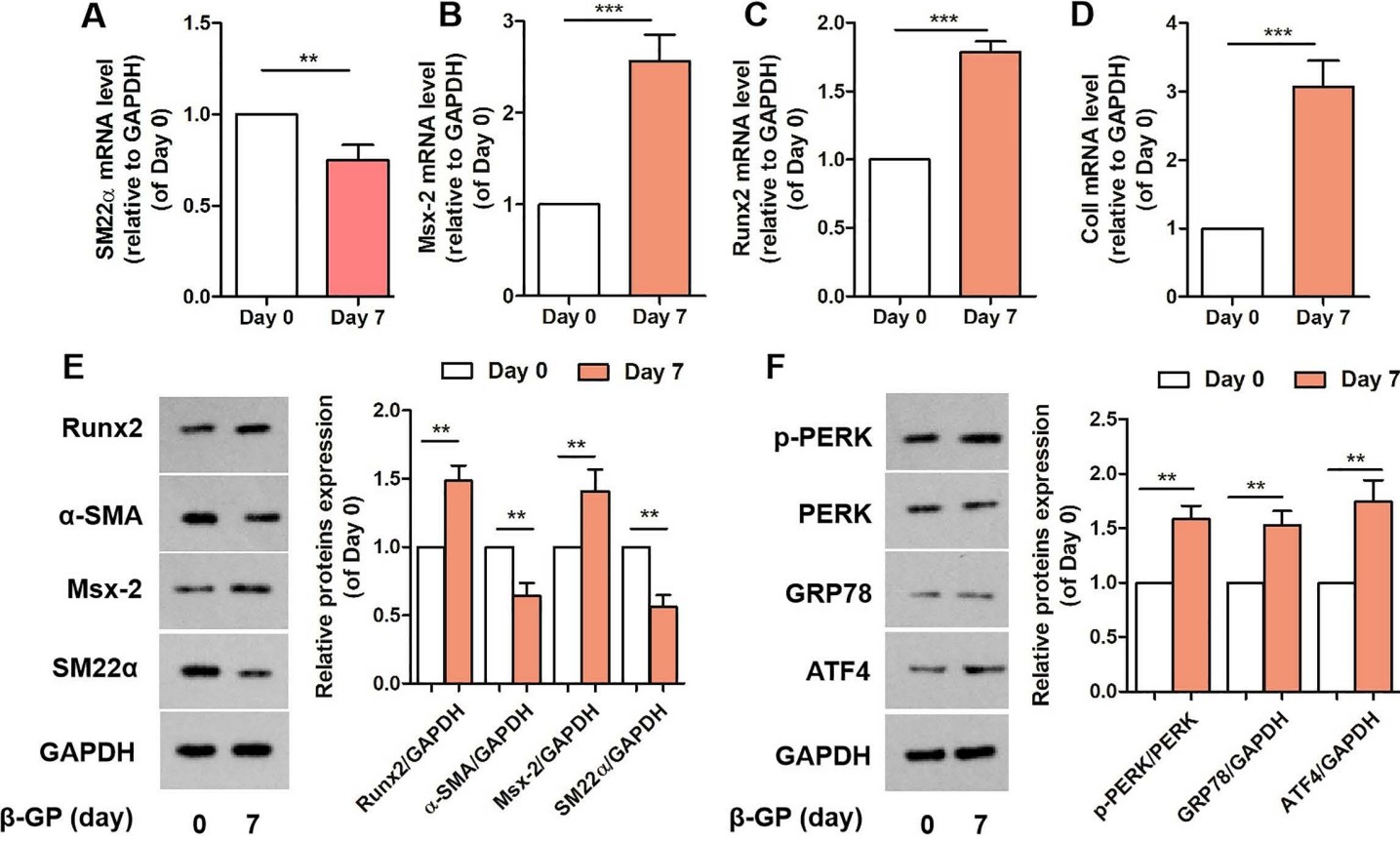

**Fig 2. Phenotypic switching and endoplasmic reticulum stress in VSMCs.** A, B, C, D. mRNA levels of SM22-α, Msx-2, Runx2, and CoⅡ in VSMCs (n = 3). E. Representative bands of Western blotting of Runx2, α-SMA, Msx-2 and SM22α in VSMCs (n = 3). F. Representative bands of Western blotting of p-PERK, PERK, GRP78 and ATF4 in VSMCs (n = 3). One-way analysis of variance and Tukey's post hoc test were used to compare multiple groups. *P < 0.05, **P < 0.01, ***P < 0.001 vs. Control.

## 4-PBA Attenuated β-glycerophosphate-induced ERS and phenotypic transition in VSMCs

To confirm the involvement of phenotypic switching and ERS in the β-glycerophosphate-induced osteogenic differentiation of VSMCs, we tested the effects of the ERS inhibitor 4-PBA on β-glycerophosphate-induced calcification. In the presence or absence of 4-PBA for 7 days, 4-PBA treatment reversed the β-glycerophosphate-induced upregulation of p-PERK, GRP78, and ATF4 (Fig 3A). Moreover, 4-PBA inhibited the expression of osteogenic differentiation-related proteins and normalized the decreased contractile markers in β-glycerophosphate-induced VSMCs, further delaying the phenotypic switching of VSMCs *in vitro* (Fig 3B). These findings indicate that 4-PBA treatment ameliorated β-glycerophosphate-induced calcification by inhibiting the switching of contractile VSMCs to an osteoblast-like phenotype.

## siSENCR reversed the β-glycerophosphate-induced VC by suppressing ERS in VSMCs

To investigate whether SENCR knockdown modulates β-glycerophosphate-mediated calcification in VSMCs, we examined the effects of SENCR knockdown on β-glycerophosphate-mediated calcification. As indicated in Fig 4, compared with negative control, the transfection efficiency of siSENCR was approximately 50%. These findings demonstrated that siSENCR transfection was successful (Fig 4A). Alizarin red S staining and cell scratch repair assays also showed that SENCR knockdown caused calcium deposition and migration in β-glycerophosphate-induced VSMCs (Fig 4B and C).

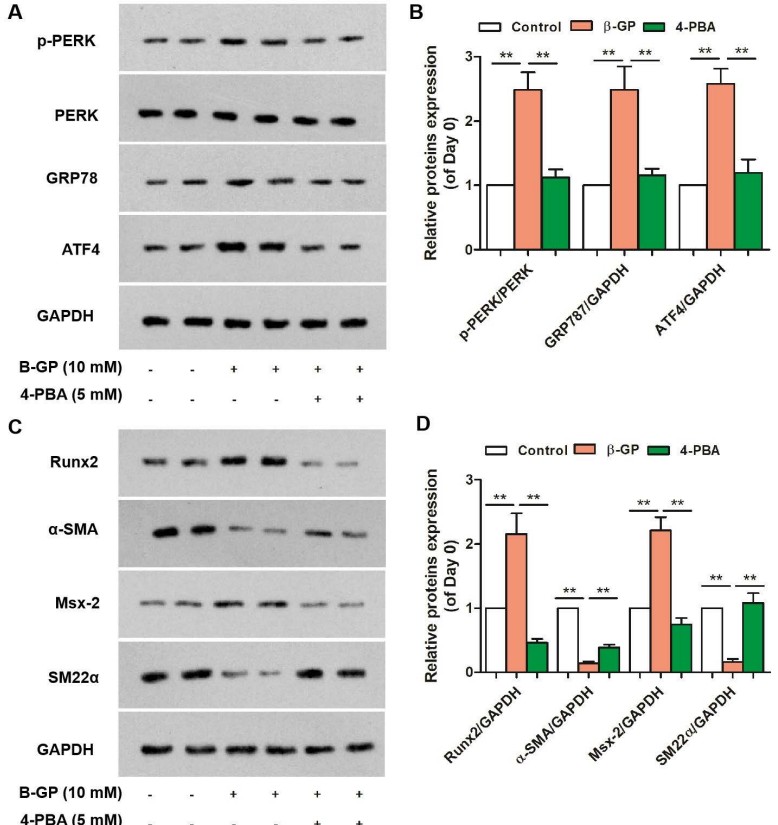

**Fig 3. 4.PBA attenuated β-glycerophosphate-induced phenotypic transition and ERS in VSMCs.** A and B. Protein expression levels of p-PERK, PERK, GRP78, and ATF4 in the 4-PBA-treated group (n = 3). C and D. Representative bands from western blotting for Runx2, α-SMA, Msx-2, and SM22-α in groups (n = 3). One-way analysis of variance and Tukey's post hoc test were used to compare multiple groups. *P < 0.05, **P < 0.01 vs. Control.

We observed the effects of SENCR knockdown on Runx2, Msx-2, and SM22-α expression. SENCR knockdown down-regulated the mRNA expression of Runx2 and Msx-2 and showed an increase in SM22-α in siSENCR-transfected cells (Fig 4D). We further assessed the protein expression of Runx2, Msx-2, α-SMA, SM22-α, p-PERK, GRP78, and ATF4 in siSENCR-transfected cells, and found that the expression of Runx2, Msx-2, p-PERK, GRP78, and ATF4 decreased. In contrast, α-SMA and SM22-α levels increased in siSENCR-transfected cells (Fig 4E, F). These findings suggest that SENCR knockdown inhibits β-glycerophosphate-induced calcification by inhibiting ERS in the VSMCs.

## SENCR promoted the ERS-mediated VC by inhibiting miR-4731-5p expression

To further clarify the role of SENCR in ERS-mediated calcification of VSMCs, bioinformatic databases for possible targeted genes related to SENCR were predicted. TargetScan results indicated that SENCR has a bindable region with miR-4731-5p, and that miR-4731-5p has a bindable region with ATF4, as predicted by bioinformatics. Therefore, a follow-up study was conducted to confirm that miR-4731-5p is downregulated in β-glycerophosphate-induced VSMCs and that intervention with SENCR negatively regulates the expression of miR-4731-5p. We further demonstrated that miR-4731-5p analogs or inhibitors suppress and activate ERS, respectively, and experimentally demonstrated that miR-4731-5p has the most significant effect on p-PERK and ATF4, thus participating in the osteogenic phenotypic switching and calcification of VSMCs.

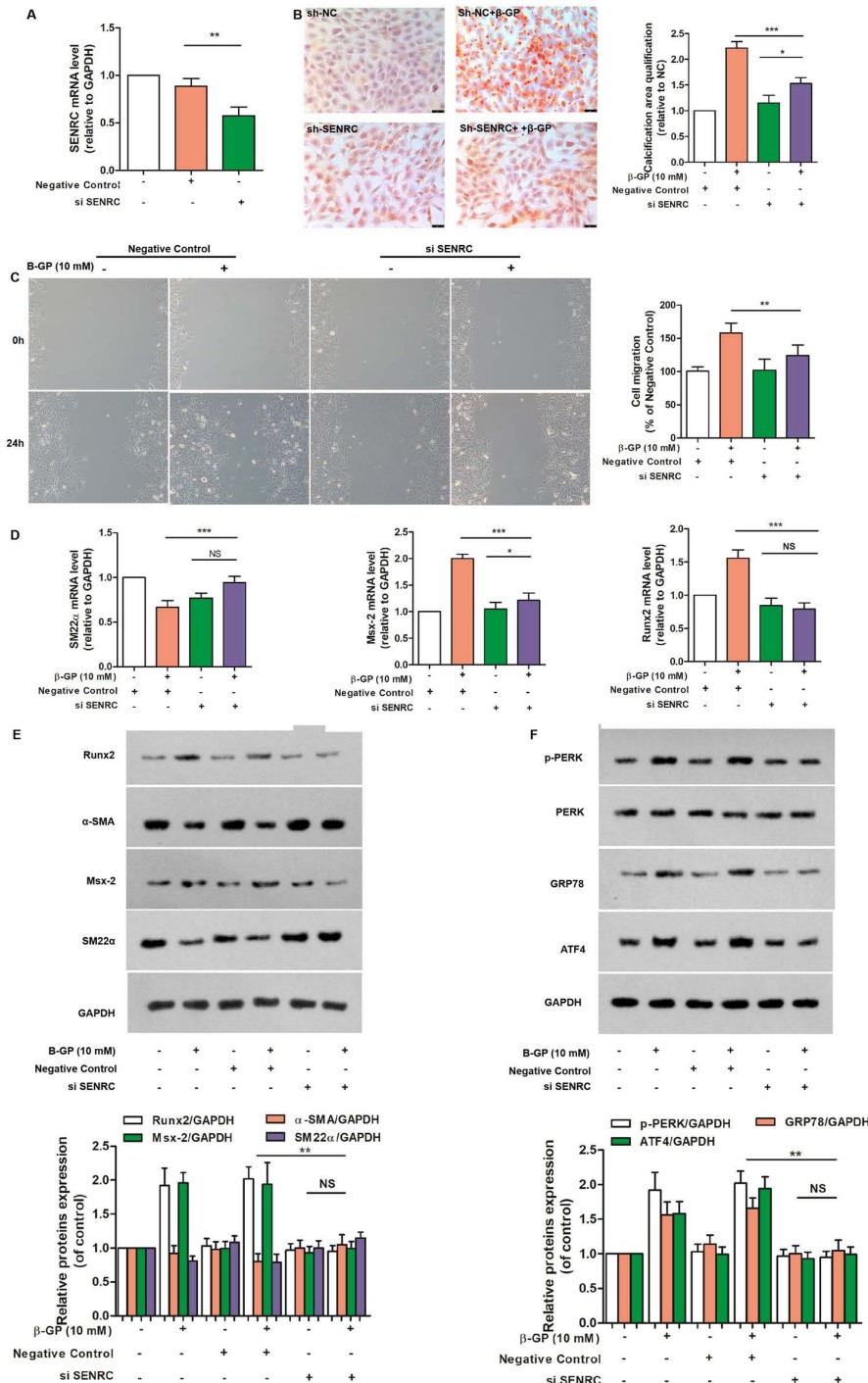

**Fig 4. si-SENRC reversed β-glycerophosphate-mediated calcification in VSMCs by inhibiting ERS in vitro.** A. Transfection efficiency of SENRC small interfering RNA lentivirus in VSMCs. Alizarin red S staining (B) and cell scratch repair assays (C) were measured to quantify the calcified areas and cell migration ability of VSMCs transfected with sh-SENRC relative to sh-NC group. D. mRNA expression levels of SM22-α, Msx-2, and Runx2 in VSMCs transfected with sh-SENRC (n = 3). E. Representative bands of Western blotting of Runx2, α-SMA, Msx-2, and SM22-α (n = 3). F. Representative bands of Western blotting of p-PERK, PERK, GRP78, and ATF4 in sh-SENRC group (n = 3). One-way analysis of variance and Tukey's post hoc test were used to compare multiple groups. *P < 0.05, **P < 0.01, ***P < 0.001.

Considering the number of binding sites, miR-4731-5p, which targets the key transcription factor ATF4 in ERS, was highly correlated with SENCR. As shown in Fig 5, miR-4731-5p expression was significantly downregulated in calcified VSMCs. SENCR knockdown significantly upregulated miR-4731–5pexpression in VSMCs, whereas SENCR

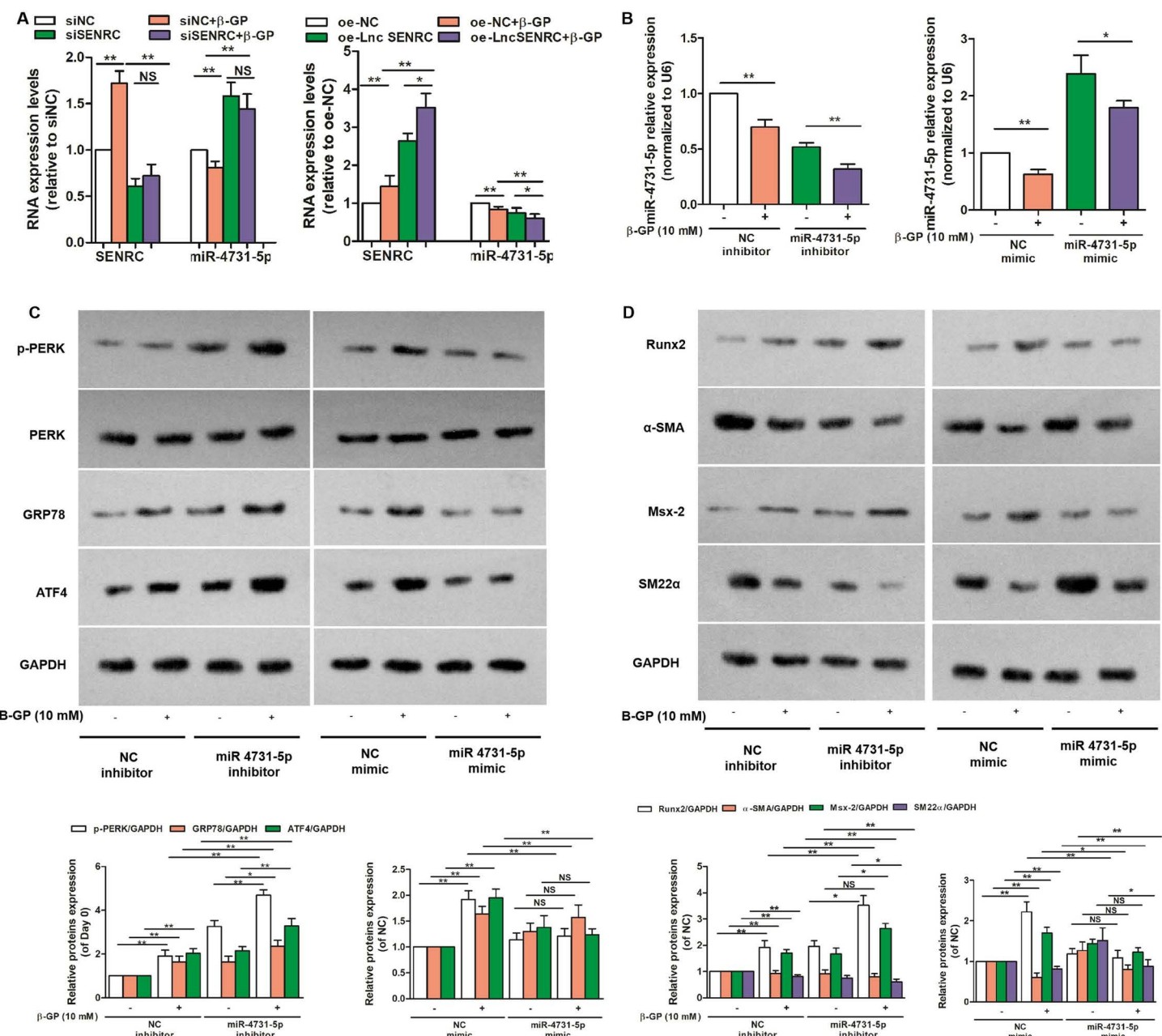

**Fig 5. SENRC promoted the ERS-mediated calcification by suppressing miR-4731-5p expression in VSMCs.** A. Effect of SENRC knockdown or overexpression on miR-4731-5p expression (n = 3); B. miR-4731-5p expression level by RT-qPCR after treatment with miR-4731-5p inhibitor or mimic (n = 3). C and D. Representative bands of Western blotting of cellular ERS-related and phenotypic transformation-related proteins in VSMCs after treatment with miR-4731-5p inhibitor or mimic (n = 3). One-way analysis of variance and Tukey's post hoc test were used to compare multiple groups. *P < 0.05, **P < 0.01, NS: no significance.

overexpression exhibited a counterproductive effect, indicating that miR-4731-5p expression was negatively correlated with SENCR (Fig 5A) and suggesting that miR-4731-5p is mediated by SENCR in β-glycerophosphate-treated VSMCs. The expression of miR-4731-5p in VSMCs was prominently inhibited or overexpressed using an miR-4731-5p inhibitor or mimic, respectively, to elucidate whether miR-4731-5p is involved in VSMC calcification (Fig 5B). As shown in Fig 5C and D, the miR-4731-5p inhibitor significantly increased the β-glycerophosphate-induced expressions of p-PERK, GRP78, and ATF4 and further activated osteogenic phenotypic transformation, whereas the miR-4731-5p mimic exerted the opposite effects. These results indicate that SENCR mediate β-glycerophosphate-induced calcification of VSMCs by downregulating miR-4731-5p to activate the PERK-GRP78-ATF4 pathway in ERS.

## Discussion

Vascular calcification occurs frequently in CKD, with high morbidity and mortality rates, and it has brought enormous economic and health-related pressure worldwide. There is mounting evidence that no effective therapy is available due to the lack of fully elucidating the mechanisms underlying VC [12]. There have been reported suggests that lncRNAs are extensively accepted as important contributors in the involvement about the pathological process of VC [13]. As previously reported, epigenetic upregulation of H19 favoured SAHH deficiency-promoted Runx2-dependent calcification *in vivo* [14]. In high phosphorus-mediated vascular calcification model rats and VSMCs, downregulation of H19 or Toll-like receptor 3 (TLR3) alleviated, whereas downregulation of miR-138 aggravated the calcification and vascular dysfunction [15]. In valve interstitial cells, tumor stemness index (TSI) alleviates aortic valve calcification by inhibiting the TGF-β/Smad3 pathway by downregulating osteoblastic differentiation [16]. Moreover, Lrrc75a-as1 acts as a negative regulator of vascular calcification and its overexpression inhibits calcium accumulation in A10 cells [17]. Several studies have demonstrated that lncRNAs are located in the nucleus and cytoplasm, and their subcellular localization patterns support the fact that lncRNAs exert multiple functions and are involved in the development of vascular calcification [18,19]. Such as, lncRNA anti-differentiation non-coding RNA (ANCR) inhibits β-glycerophosphate-induced osteoblastic differentiation and mineralization in VSMCs [20]. It is well established that β-glycerophosphate can alter vascular function via genomic mechanisms, in which β-GP-mediated vascular injury to modulate the vascular gene expression in VSMCs [21]. Recent studies have further indicated the role of SENCR in accelerating the osteogenic differentiation and calcification of VSMCs [22,23]. The present study revealed two novel findings: [1] excess β-glycerophosphate increased SENCR levels by activating ERS, and [2] Inhibition of SENCR suppressed osteogenic differentiation of VSMCs mediated ERS-dependently via sponging miR-4731-5p.

Increasing evidence indicates lncRNAs are widely accepted as important regulators of gene expression and exert diverse of biological functions. Recently, the involvement of associated lncRNAs in vascular calcification, has attracted special interest. Several studies have demonstrated that metastasis-associated lung adenocarcinoma transcript-1 (MALAT1) promotes the vascular calcification, and knockdown of MALAT1 suppresses VSMCs calcification [24]. Another lncRNA ANCR, inhibits β-glycerophosphate-induced conversion and mineralization of VSMCs [25]. SENCR is a vascular-specific lncRNA that regulates the differentiation state and contractile phenotype of muscle cells [26]. Emerging evidence indicates that the functional features of SENCR in atherosclerosis are the basis of VC, demonstrating its potential role in VC [27]. The present findings demonstrate that SENCR expression increases in the presence of excess β-glycerophosphate, both *in vivo* and *in vitro*. SENCR knockdown significantly abrogated β-glycerophosphate-induced osteogenic differentiation and calcification in VSMCs.

Increasing evidence supports that VC is a multifactorial process of medial calcification, characterized by contractile VSMCs conversion to an osteochondrogenic phenotype [28]. The present study demonstrated that excess β-glycerophosphate stimulates the phenotypic switch of VSMCs during the procalcific process, as supported with the raises of osteogenic markers including Runx2, BMP2, and osteocalcin and the decreases in contractile markers such as α-SMA, SM22α, and smoothelin, during which Runx2 exerts a crucial role. It has been reported that Runx2 pertains to

the runt-associated transcription factor family and guards osteogenic differentiation among multiple osteogenic signals. Several studies have demonstrated that Runx2 is expressed at low levels in normal vascular cells, but is over-activated in calcified vascular tissue, especially in atherosclerotic plaques. It has been reported to Runx2 a pivotal role in osteoblast differentiation of VSMCs and development of β-glycerophosphate-enhanced calcification [29]. Recently, the epigenetics of Runx2 have been regarded as a prominent signaling factor impacting its stability and transcriptional activity [14,30]. Nevertheless, whether SENCR are involved in β-glycerophosphate-induced Runx2 expression and the osteoblastic differentiation of VSMCs remains unknown. LncRNAs are post-transcriptional regulators that generally function as natural miRNA sponges and interfere with miRNA pathways. Studies have revealed that lncRNAs are highly involved in medical calcifications and exerts a pro-mineralization function [15,30,31]. Accumulating evidence suggests that SENCR binds to microR-4731-5p as a sponge to inhibit downstream effects via a β-glycerophosphate-induced Runx2-dependent pathway [32–34]. In our preliminary study, we predicted the interaction between SENCR and miR-4731-5p using lncRNA microarrays, and further confirmed the existence of target sites between the two molecules. Silencing miR-4731-5p reversed the suppression of Runx2-dependent osteogenic differentiation and calcification induced by β-glycerophosphate-mediated upregulation of SENCR. Finally, we found that SENCR inhibited miR-4731-5p and that SENCR and miR-4731-5p were in the same RNA-induced silencing complex. Collectively, these findings suggest that SENCR activation is essential for β-glycerophosphate-induced Runx2-dependent osteogenic differentiation and calcification of VSMCs via sponging miR-4731-5p.

The limitations of present study were that SENCR exhibits positive effects via regulating expression of RUNX2 by targeting miR-4731-5p at the post-transcriptional level using a bioinformatics method, but we just focused on the role of SENCR/miR-4731-5p in vitro. Therefore, we cannot rule out the possibility that SENCR inhibition suppressed the vascular calcification. Secondly, we did not offer direct evidence that SENCR knockout in the presence of excess β-glycerophosphate could decrease calcification. The use of SENCR-knockout mice in the presence of excess β-glycerophosphate may provide more direct evidence for the role of SENCR in vascular calcification. Finally, a prospective study is required to further elucidate the association between SENCR expression and all-cause mortality in CKD patients. Our study offers insights into the management by which lncRNAs affect vascular calcification and contributes to the development of potential therapeutic strategies against CKD-associated vascular calcification.

## Conclusions

Collectively, the present study clearly identified SENCR as a novel positive regulator of β-glycerophosphate-induced phenotypic switching and calcium deposition in VSMCs. We have found that SENCR positively regulates the RUNX2 expression at the post-transcriptional level by sponging miR-4731-5p in the setting of excess β-glycerophosphate. Furthermore, it suggests that a probe to explore SENCR blockade may attenuate pathological vascular calcification mediated by β-glycerophosphate.

## Acknowledgments

Not applicable.

## Author contributions

**Conceptualization:** Yongpan Huang, wei Zhan.

**Formal analysis:** Chong Song.

**Funding acquisition:** Yongpan Huang.

**Investigation:** wei Zhan, Chong Song, Sina Wu.

**Methodology:** Meihua Tan.

**Resources:** Yongpan Huang, Meihua Tan.

**Software:** Meihua Tan, Li Wu.

**Supervision:** Sina Wu.

**Validation:** Yongpan Huang, Li Wu.

**Visualization:** Yongpan Huang.

**Writing – original draft:** Yongpan Huang, wei Zhan, Chong Song, Sina Wu.

**Writing – review & editing:** Yongpan Huang, wei Zhan, Chong Song, Meihua Tan, Li Wu, Sina Wu.

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
