## [Decision Letter · Decision Letter 0]

4 Dec 2024

PONE-D-24-45289LncRNA SENRC knockdown alleviates β-glycerol phosphate-mediated vascular calcification via miR-4731-5p by suppressing endoplasmic reticulum stressPLOS ONE

Dear Dr. Huang,

Thank you for submitting your manuscript to PLOS ONE. After careful consideration, we feel that it has merit but does not fully meet PLOS ONE’s publication criteria as it currently stands. Therefore, we invite you to submit a revised version of the manuscript that addresses the points raised during the review process.

The paper has the potential to be of interest and sufficiently novel but is not yet robust enough in its current format (there are some inconsistencies in the data; the statistical analysis is unclear; the number of experimental replicates is minimal). Furthermore, the presentation of the data, discussion and bibliography must be improved. Moreover it would benefit from a revision of the English style and manner of expression. 

We look forward to receiving your revised manuscript.

Kind regards,

Maria Cristina Vinci, PharmD, PhD

Academic Editor

PLOS ONE

https://journals.plos.org/plosone/s/file?id=ba62/PLOSOne_formatting_sample_title_authors_affiliations.pdf "

“This work was supported by the funding of Professor and PhD at Changsha Social Work College (2023JB24) and the Hunan Provincial Natural Science Foundation (no. 2023JJ60262).”

4. In the online submission form, you indicated that “The data underlying the results presented in the study are available from CORRESPONDING AUTHOR”

Reviewers' comments:

Reviewer's Responses to Questions

**Comments to the Author**

1. Is the manuscript technically sound, and do the data support the conclusions?

Reviewer #1: Partly

2. Has the statistical analysis been performed appropriately and rigorously? 

Reviewer #1: No

3. Have the authors made all data underlying the findings in their manuscript fully available?

Reviewer #1: No

4. Is the manuscript presented in an intelligible fashion and written in standard English?

Reviewer #1: No

5. Review Comments to the Author

Reviewer #1: In this paper, the authors investigate the mechanism underlying vascular calcification (VC) in an in vitro model consisting of VSMCs treated with β-glycerol phosphate (β-GP). Through pharmacological and molecular approaches, they found that β-GP positively regulates the expression of LncRNA SENRC in an ERS-dependent manner. They proposed that the LncRNA SENRC sponges miR-4731-5p to modulate the osteogenic differentiation of VSMCs via Runx2.

Understanding the mechanisms of vascular calcification is of paramount importance for the development of new therapies. Although ER-stress and long non-coding RNA have been involved in this process, the authors identify a novel possible SENRC/miR-4731-5p axis that has previously been shown to regulate VSMCs proliferation (https://doi.org/10.1002/jcp.29775). From this point of view the paper is sufficiently novel and relevant to the field. However, there are several points that need to be addressed.

Major points

1) The data presented are limited to an in vitro model of VC. To confirm the relevance of the SENRC/miR-4731-5p axis and propose it as a possible target for pharmacological intervention, the authors should verify the modulation of the two transcripts (SENRC/miR-4731-5p) in preclinical or clinical models (they could also analyse published data sets).

2) ENGLISH. The whole text would benefit from an English style revision. Several sentences are unclear or ambiguous. Just a few examples, lines 45-49 or 63-66, 248-250, 251-253, 253-254.

Lines 265-267 please reconsider the sentence because two opposite effects of β-GP were proposed “the study revealed two novel findings: (1) excess β-GP increased SENRC levels by activating ERS during the procalcific process, and (2) β-GP-mediated downregulation of SENRC promoted the ERS-dependent osteogenic differentiation of VSMCs by sponging miR-4731-5p”.

Lines 240-242 please reconsider the sentence. The data description refers to fig 5C and D but not to fig 5B

3) DISCUSSION. Reconsider the discussion. There is often repetition in the text. For example, lines 271-274/276/277

The first part of the discussion is almost an introduction

The Limitation and conclusion section is unclear, please reformulate

4) REFERENCES. Some sentences are not supported by the relevant literature. Just a few examples:

Lines 79-81 “Given the close association between excess β-glycerol phosphate, SENRC, ERS, and VC,”

Other examples lines 263-265/267-269/280-281/282

Lines 306-307 Accumulating evidence suggests that SENRC binds to microR-4731-5p as a sponge to inhibit downstream effects via a β-GP-induced Runx2-dependent pathway. This sentence is very relevant to the paper, but there is no relative reference

Reference 30 does not refer to LncSENRC

Although the SENRC/miR-4731-5p axis has been implicated for the first time in VC, the functional interaction between SENRC and miR-4731-5p has already been predicted by https://doi.org/10.1016/j.prp.2023.154483. Not cited in the text.

5) DATA ANALYSIS AND PRESENTATION. Often the number of experimental replicates is 3/4, please report the individual values in the bars (at least for Figures 1-3).

Please indicate in each figure legend the type of statistical analysis performed (t-test, two-way ANOVA, etc.)

Representative blots do not always correspond to their quantification, especially those involving Gpr78.

Figure 2. There is an apparent inconsistency between mRNA and protein levels for Sm22. Sm22 increases as mRNA but decreases as protein. This data should be discussed.

6) PRESENTATION OF THE FIGURES. Figures 4 and 5 are very complex. To avoid confusion, please report the letter in the upper left corner of each panel. Also in figure 4, report the title for the 4 pictures in panel 4B

Minor points

1) Please provide the complete image of the representative blots in the supplementary materials for referees. It would be preferable to present a couple of samples, rather than just one, as shown.

2) Abbreviations: Please define the acronyms on first mention within the text. Additionally, several acronyms remain unidentified (SENCR, TS1, etc.).

VC is vascular calcification, not vascular calcium (abstract row 24).

3) Please ensure that the correct nomenclature of genes and proteins is used.

4) Row 200: This is probably fig 3A.

6. PLOS authors have the option to publish the peer review history of their article (what does this mean? ). If published, this will include your full peer review and any attached files.

**Do you want your identity to be public for this peer review?** For information about this choice, including consent withdrawal, please see our Privacy Policy .

Reviewer #1: No

---

## [Author Response · Author response to Decision Letter 1]

9 Feb 2025

Responses to Reviewers' comments:

Major points

1) The data presented are limited to an in vitro model of VC. To confirm the relevance of the SENRC/miR-4731-5p axis and propose it as a possible target for pharmacological intervention, the authors should verify the modulation of the two transcripts (SENRC/miR-4731-5p) in preclinical or clinical models (they could also analyse published data sets).

Answer Thank you for your suggestions. Previous studies have demonstrated that overexpression of SENRC attenuated the proliferation, migration and phenotypic switching of vascular smooth muscle cells in AD patients and f atherosclerosis (AS). Furthermore, SENRC alleviated hypoxia/reoxygenation-induced cardiomyocyte apoptosis and inflammatory response in acute myocardial infarction. Especially, SENRC has been reported that its regulation in the malignant phenotype of AML cells by targeting miR-4731-5p. However, whether SENRC/miR-4731-5p involved in regulating osteogenic differentiation of VSMCs and VC remained unclear.

References:

1.Yi Song, Tao Wang, Chunjie Mu, Wenting Gui, Yao Deng, Runwei Ma.LncRNA SENCR overexpression attenuated the proliferation, migration and phenotypic switching of vascular smooth muscle cells in aortic dissection via the miR-206/myocardin axis.Nutr Metab Cardiovasc Dis. 2022 Jun;32(6):1560-1570.

2.M. Chen, Y. Guo, Z. Sun, et al. Long non-coding RNA SENCR alleviates hypoxia/reoxygenation-induced cardiomyocyte apoptosis and inflammatory response by sponging miR-1.Cardiovasc Diagn. Ther., vol. 11 (3) (2021), pp. 707-715.

3.F. Ye, J. Zhang, Q. Zhang, et al. Preliminary study on the mechanism of long noncoding RNA SENCR regulating the proliferation and migration of vascular smooth muscle cells. J. Cell Physiol., vol. 235 (12) (2020), pp. 9635-9643.

4.Q. Lyu, S. Xu, Y. Lyu, et al. SENCR stabilizes vascular endothelial cell adherens junctions through interaction with CKAP4.Proc. Natl. Acad. Sci., vol. 116 (2) (2019), pp. 546-555.

5.Changhao Han, Yan Qi, Yuanting She, et al,.Long noncoding RNA SENCR facilitates the progression of acute myeloid leukemia through the miR-4731-5p/IRF2 pathway.Pathol Res Pract. 2023 May:245:154483.

2) ENGLISH. The whole text would benefit from an English style revision. Several sentences are unclear or ambiguous. Just a few examples, lines 45-49 or 63-66, 248-250, 251-253, 253-254.

Answer Thank you for your suggestions. We have checked and revised in the manuscript. Thank you for bettering the quality about our manuscript.

Lines 265-267 please reconsider the sentence because two opposite effects of β-GP were proposed “the study revealed two novel findings: (1) excess β-GP increased SENRC levels by activating ERS during the procalcific process, and (2) β-GP-mediated downregulation of SENRC promoted the ERS-dependent osteogenic differentiation of VSMCs by sponging miR-4731-5p”.

Answer Thank you for your suggestions. We have checked and revised in the manuscript. Thank you for bettering the quality about our manuscript.

Lines 240-242 please reconsider the sentence. The data description refers to fig 5C and D but not to fig 5B

Answer Thank you for your suggestions. We have checked the sentence and added the data descriptions about the fig 5B, fig 5C and fig 5D in the manuscript. Thank you for bettering the quality about our manuscript.

3) DISCUSSION. Reconsider the discussion. There is often repetition in the text. For example, lines 271-274/276/277

The first part of the discussion is almost an introduction

The Limitation and conclusion section is unclear, please reformulate

Answer Thank you for your suggestions. We have checked the possible duplicate parts and deleted those in order to integrity in the manuscript. Furthermore, we have reformulated in these parts. Thank you for bettering the quality about our manuscript.

4) REFERENCES. Some sentences are not supported by the relevant literature. Just a few examples:

Lines 79-81 “Given the close association between excess β-glycerol phosphate, SENRC, ERS, and VC,”

Other examples lines 263-265/267-269/280-281/282

Lines 306-307 Accumulating evidence suggests that SENRC binds to microR-4731-5p as a sponge to inhibit downstream effects via a β-GP-induced Runx2-dependent pathway. This sentence is very relevant to the paper, but there is no relative reference

Reference 30 does not refer to LncSENRC

Although the SENRC/miR-4731-5p axis has been implicated for the first time in VC, the functional interaction between SENRC and miR-4731-5p has already been predicted by https://doi.org/10.1016/j.prp.2023.154483. Not cited in the text.

Answer Thank you for your suggestions. According to reviewers’ suggestions, we have checked and cited the proper and relative references in the revised manuscript.

5) DATA ANALYSIS AND PRESENTATION. Often the number of experimental replicates is 3/4, please report the individual values in the bars (at least for Figures 1-3).

Please indicate in each figure legend the type of statistical analysis performed (t-test, two-way ANOVA, etc.)

Representative blots do not always correspond to their quantification, especially those involving Gpr78.

Figure 2. There is an apparent inconsistency between mRNA and protein levels for Sm22. Sm22 increases as mRNA but decreases as protein. This data should be discussed.

Answer Thank you for your suggestions.

In the section of DATA ANALYSIS AND PRESENTATION, the number of experimental replicates is 3 times and the values of the bar were ratios between the statistic values.

We have indicated the type of statistical analysis performed in the manuscript.

According to the blots about their quantification, especially those involving Gpr78, the values were ratios compared with control groups. We think there were no debates about the values. Furthermore we have provided all the bots for the experiments.

In figure 2, in the process of data statistics, we made some mistakes about the inconsistency between mRNA and protein levels, actually the changes in mRNA and protein levels are consistent. Sorry for disturbing you about confusion.

6) PRESENTATION OF THE FIGURES. Figures 4 and 5 are very complex. To avoid confusion, please report the letter in the upper left corner of each panel. Also in figure 4, report the title for the 4 pictures in panel 4B

Answer Thank you for your suggestions. We have revised and represent the Figures in the revised manuscript including all the Figure 1,2,3,4,5. Thank you for bettering the quality about our manuscript.

Minor points

1) Please provide the complete image of the representative blots in the supplementary materials for referees. It would be preferable to present a couple of samples, rather than just one, as shown.

Answer Thank you for your suggestions. We have provided the the complete image of the representative blots in the supplementary materials for referees

2) Abbreviations: Please define the acronyms on first mention within the text. Additionally, several acronyms remain unidentified (SENCR, TS1, etc.).

Answer Thank you for your suggestions. We have checked in the manuscript.

SENCR:

Smooth muscle and endothelial cell-enriched migration/differentiation-associated lncRNA

TS1:tumor stemness index (TSI); ANCR: lncRNA anti-differentiation noncoding RNA (ANCR)

VC is vascular calcification, not vascular calcium (abstract row 24).

Answer Thank you for your suggestions. We have checked and revised in the manuscript.

3) Please ensure that the correct nomenclature of genes and proteins is used.

Answer Thank you for your suggestions. We have checked the correct nomenclature of genes and proteins.

4) Row 200: This is probably fig 3A.

Answer Thank you for your suggestions. We have revised the error in Row in the manuscript. The figure is actually figure 3A

---

## [Editor Report · Decision Letter 1]

1 Apr 2025

The lncRNA SENCR knockdown alleviates vascular calcification via miR-4731-5p by suppressing endoplasmic reticulum stress

PONE-D-24-45289R1

Dear Dr. Yongpan Huang,

We’re pleased to inform you that your manuscript has been judged scientifically suitable for publication and will be formally accepted for publication once it meets all outstanding technical requirements.

Kind regards,

Maria Cristina Vinci, PharmD, PhD

Academic Editor

PLOS ONE

---

## [Editor Report · Acceptance letter]

PONE-D-24-45289R1

PLOS ONE

Dear Dr. Huang,

I'm pleased to inform you that your manuscript has been deemed suitable for publication in PLOS ONE. Congratulations! Your manuscript is now being handed over to our production team.

Kind regards,

on behalf of

Dr Maria Cristina Vinci

Academic Editor

PLOS ONE